# CompassJudger-2: A Holistic Approach Towards Generalist Judge Model

## Abstract

Recently, the role of LLM-as-judge in evaluating large language models has gained prominence, emerging as an important method to partially replace costly human assessment. However, current judge models suffer from narrow specialization and limited robustness, undermining their capacity for comprehensive evaluations. In this work, we decompose the ability of a generalist and generative judge model into three levels: objective verification, subjective evaluation, and rubric refinement. We present CompassJudger-2, a novel generalist judge model that overcomes these limitations via a task-driven, multi-scenarios data curation strategy. We conducted large-scale data collection for each type of task and designed tailored rejection sampling strategies to filter the data, ensuring data diversity, accuracy, and effectiveness. Empirically, CompassJudger-2 achieves superior results across multiple judge and reward benchmarks, demostrating its excellent robustness and generalization ability. These contributions advance robust, scalable LLM judgment and establish new performance and evaluation standards.

## 1 Introduction

In recent years, large language models (LLMs) have advanced rapidly with the development of new foundation models such as DeepSeek-R1 (Guo et al., 2025), GPT series (OpenAI, 2025), and the Qwen series (Yang et al., 2024). Innovations in architecture and data scaling have enabled LLMs to achieve state-of-the-art performance across diverse tasks, including natural language understanding, code generation, creative writing, and complex reasoning (Li et al., 2024; Jain et al., 2024; Dubois et al., 2024; Lambert et al., 2024).

As large language models (LLMs) become increasingly integrated into real-world applications, accurately evaluating the quality of their responses has become more important than ever. Rule-based methods (Rein et al., 2024; Zhou et al., 2023; Jain et al., 2024; Clark et al., 2019; Contributors, 2023; Dua et al., 2019) perform well on standardized tasks with clearly defined ground truth. However, These methods typically rely on complex regular expressions, which limit them to objective answers. As a result, they are fragile in edge cases and ineffective when dealing with open-ended questions. To address these challenges, model-based evaluation methods known as LLM-as-a-Judge (Zhu et al., 2023; Chen et al., 2025a; Ye et al., 2024; Yu et al., 2025) have been proposed. These approaches leverage the reasoning capabilities of LLMs to automate the evaluation pipeline and reduce the need for human intervention. However, the broader adoption of such methods is constrained by the limited generalization abilities of current judge models. Existing models are usually trained on specific tasks and domains due to the lack of diverse training data and incomplete world knowledge. Consequently, they may produce invalid or even harmful analyses and explanations when applied to unfamiliar tasks or domains, which poses a barrier to the continued improvement and evolution of LLMs.

Guided by Sadler's formative assessment theory (Sadler, 1989), which characterizes effective evaluation as a progression from determining correctness, to appraising the quality of open-ended responses, and ultimately to providing improvement-oriented feedback, we break down the judging capability into three progressively more advanced stages.First, an effective judge model should be able to assess the correctness of a candidate response against a reference answer. This ability is especially important in objective scenarios where ground truth is clearly defined. Although current RLVR pipelines often rely on rule-based reward systems, an ideal judge model would possess the reasoning capacity to interpret reference answers intelligently, avoiding incorrect rejections of valid responses that merely

Figure 1: **Decomposition of Different Judging Abilities.** We decompose the essential capabilities of a generalist judge into objective verification, subjective evaluation, and rubric refinement.

differ in form. Second, the judge model should also function effectively in the absence of a reference answer. This is crucial for subjective and open-ended questions, where no single correct answer exists. In such instances, the model must analyze the content of responses and conduct comparative evaluations across multiple answers to identify which one is more appropriate. At a more advanced level, a capable judge model should provide constructive feedback aimed at improving candidate responses. This is best achieved through fine-grained, rubric-based evaluations. To fulfill this role, the model must develop a deep understanding of both the question and the response, allowing it to assess their alignment according to decomposed rubric criteria and offer detailed suggestions that meaningfully enhance the original answer.

In this paper, we establish a systematic framework for developing the generalist judge models defined above, in case of training pipeline and evaluation benchmarks. Specifically, we design distinct data construction pipelines for three different scenarios: objective verification, subjective evaluation, and rubric refinement. To more effectively evaluate the capabilities of subjective evaluation and rubric refinement, we introduce JudgerBenchV2 and RubricBench, paving the way for standardized judge evaluation. For training data construction, we adopt rejection sampling and a delta-based strategy for rubric generation to enhance data quality. Furthermore, by incorporating novel verifiable judge rewards into the supervised fine-tuning (SFT) process, we obtain CompassJudger-2, which demonstrates strong performance across a wide range of judging benchmarks.

To summarize, our contributions are as follows:

- We decompose judge ability into three different scenarios: objective verification, subjective evaluation and rubric refinement.
- We introduce JudgerBenchV2, which treats a Mix-of-Judgers as the ground truth and deploys new metrics that jointly assess accuracy and rank fidelity, enabling more reliable evaluation. We also introduce RubricBench to standardize the evaluation of rubric refinement.
- We develop a judge model named CompassJudger-2, which significantly improves judging performance across various scenarios, paving the way toward a generalist judge.

## 2 RELATED WORKS

**Objective Verifiers.** Verifiers are typically developed to assess objective questions with standard answers, and their outputs usually involve hard labels such as "correct" or "incorrect" (Cobbe et al., 2021; Kawabata & Sugawara, 2024). Mainstream approaches often employ pattern matching (e.g., extracting phrases like "The answer is []") (Contributors, 2023; Gao et al., 2024; OpenAI, 2023) or utilize specialized tools for mathematical expressions (e.g., Math-Verify (huggingface, 2024)). However, due to the unstructured nature of generative model outputs, such character-level matching methods are prone to failure. As LLMs become increasingly powerful, researchers have begun exploring the use of general LLMs or developing specialized verifiers to achieve more robust answer matching (Chen et al., 2025a; Ma et al., 2025; Su et al., 2025). In our work, we treat answer matching, particularly outcome verification, as a component of CompassJudger-2. In contrast to these task-specific verifiers, our model is capable of following more diverse prompts, thereby enabling more robust and versatile answer verification capabilities.

**Subjective Judgers.** LLM judgers can play a role in providing judgments on model responses for open-ended questions. Unlike traditional reward models that output a single scalar value, LLM-based judges can provide richer feedback by explaining the logic and rationale underlying their

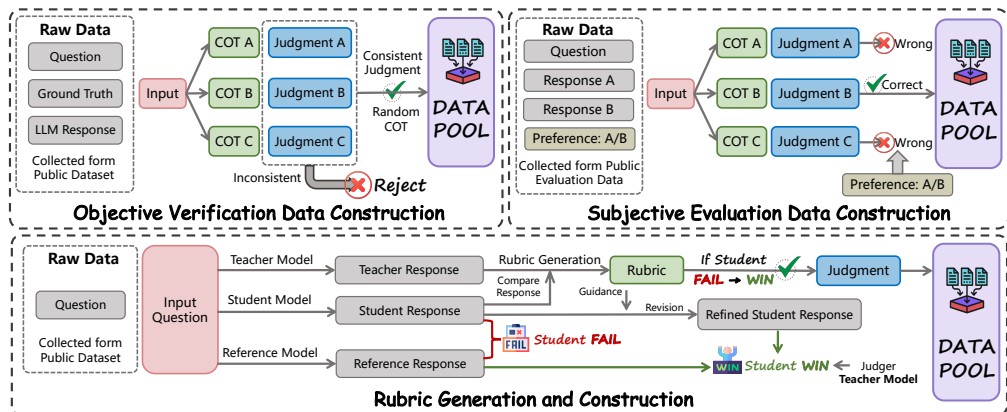

Figure 2: **Pipeline of Data Construction.** We collect data from public datasets and employ rejection sampling to filter high-quality training data across the three scenarios.

evaluations. However, many existing judge models (Zhu et al., 2023; Li et al., 2023b) are trained on specific prompts, leading to poor generalization and limited adaptability to varied evaluation scenarios. Therefore, all-in-one generative models have emerged, with CompassJudger-1 (Cao et al., 2024) being the first to incorporate a wide range of judge tasks into model training, greatly enhancing the generalization ability. Con-J (Ye et al., 2024) and RISE (Yu et al., 2025) have also conducted all-in-one Judge model training and achieved better Judge performance through the DPO strategy.

**Rubric Refiners.**  Recently, rubric-based rewards have gained significant attention due to their ability to decompose evaluation into more interpretable components. A rubric consists of questions or checkpoints focused on specific criteria for each task. In general, rubrics provide a more fine-grained basis for judgment, and many recent works have employed them as reward signals in RLVR. Rubicon-Preview (Huang et al., 2025) introduces a framework that leverages rubric-based rewards within reinforcement learning to enhance large language model performance on subjective, open-ended tasks. RaR (Gunjal et al., 2025) employs structured checklists as reward signals to train language models on complex tasks where correctness is subjective and not easily verifiable.

## 3 METHODOLOGY

### 3.1 LLM AS OBJECTIVE VERIFIER

#### 3.1.1 DATA PIPELINE

**Data Curation.**  We aggregated responses from multiple models (e.g., Qwen3-Series (Yang et al., 2025), DeepSeek-Series (Guo et al., 2025), GPT-Series (OpenAI, 2025), LLaMa-Series (Touvron et al., 2023)) across various datasets (e.g., MMLU(Wang et al., 2024), CMMLU (Li et al., 2023a), GSM8K (Cobbe et al., 2021)) and utilized DeepSeek-V3 and Qwen2.5-3B-Instruct to verify these responses, determining their correctness. We selected datasets based on their domain and question types, while employing models of varying scales to generate responses, aiming to maximize the diversity of the answer verification data.

**Rejection Sampling.**  Since the golden verification labels for answer verification tasks require annotations from human experts, it is challenging to construct and synthesize large-scale training data. To address this, we employed consistency-based rejection sampling to enhance data difficulty while maintaining accuracy. Specifically, we used both DeepSeek-V3 and the Qwen2.5-3B-Instruct model to perform multiple verifications of the same LLM's response against the ground truth. We then selected data where DeepSeek-V3 consistently provided the same verification outcome across multiple trials, while Qwen2.5-3B-Instruct showed inconsistency, thereby implementing rejection sampling for the answer verification task. For the multiple CoT trajectories synthesized by DeepSeek-V3, we randomly selected one trajectory per raw data sample to add to the training data pool, thereby ensuring the diversity of the training data.

Table 1: **Total loss and mapping functions.** We discuss three mapping functions to approximate $g$.

| Loss | Loss Function |
|---|---|
| Total Loss | $\mathcal{L}_{total} = -\frac{1}{NM}\sum_{i=1}^{N}\sum_{j=1}^{M}[\sum_{t\neq k_{ij}}\log\pi_\theta(y_t^{(i,j)}|x^{(j)},y_{<t}^{(i,j)}) + g\left(\log\pi_\theta(y_{k_{ij}}^{(i,*)}|x^{(j)},y_{<k_{ij}}^{(i,j)})\right)]$ |
| DPO Mapping Loss | $\mathcal{L}_{\text{DPO}} = -\frac{1}{NM}\sum_{i=1}^{N}\sum_{j=1}^{M}\log\sigma\left(\beta\log\frac{\pi_\theta(y_{k_{ij}}^{(i,*)}|x^{(j)},y_{<k_{ij}}^{(i,j)})}{\pi_\theta(y_{k_{ij}}^{(i,-)}|x^{(j)},y_{<k_{ij}}^{(i,j)})}\right)$ |
| Temperature Mapping Loss | $\mathcal{L}_{\text{Temp}} = -\frac{1}{NM}\sum_{i=1}^{N}\sum_{j=1}^{M}\log\frac{\exp(\log\pi_\theta(y_{k_{ij}}^{(i,*)}|x^{(j)},y_{<k_{ij}}^{(i,j)})/\tau)}{\sum_{y'}\exp(\log\pi_\theta(y'|x^{(j)},y_{<k_{ij}}^{(i,j)})/\tau)}$ |
| Margin Mapping Loss | $\mathcal{L}_{\text{Margin}} = \frac{1}{NM}\sum_{i=1}^{N}\sum_{j=1}^{M}\max\left(0,\gamma-\log\pi_\theta(y_{k_{ij}}^{(i,*)}|x^{(j)},y_{<k_{ij}}^{(i,j)})+\log\pi_\theta(y_{k_{ij}}^{(i,-)}|x^{(j)},y_{<k_{ij}}^{(i,j)})\right)$ |

## 3.2 LLM as Subjective Evaluator

### 3.2.1 Incorporating Verified Reward

**Judge Reward.** In the judge task, the model performs binary classification by outputting its prediction at designated positions, as shown in Fig.2. This structured output enables us to utilize the ground truth labels as explicit guidance signals for optimization. Inspired by DeepSeek-R1 (Guo et al., 2025), given an instruction-response pair $(x, y)$, a prediction position $k_x$, and the corresponding ground truth label $y_{k_x}^*$, we apply a rule-based reward $r(x, y)$ defined as 1 if the model's prediction at position $k_x$ matches the ground truth label $y_{k_x}^*$, and 0 otherwise.

**Policy Gradient Optimization.** We formulate the learning objective as maximizing the expected reward over the response distribution and the gradient of this objective can be derived as follows:

$$\begin{aligned}
\nabla_\theta J(\theta) &= \nabla_\theta\left[\mathbb{E}_{x\sim D}\mathbb{E}_{y\sim\pi_\theta}[r(x,y)]\right] \\
&= \mathbb{E}_{x\sim D}\mathbb{E}_{y\sim\pi_\theta}\left[r(x,y)\nabla_\theta\log\pi_\theta(y|x)\right] \\
&= \mathbb{E}_{x\sim D}\mathbb{E}_{y\sim\pi_\theta}\left[r(x,y)\sum_{t=1}^{n}\nabla_\theta\log\pi_\theta(y_t|x,y_{<t})\right]
\end{aligned} \tag{1}$$

This decomposition shows how the gradient propagates through all sequence positions in autoregressive models. Given that reward function only depends on the prediction at position $k_x$, the Judge-PG loss can be further obtained as follows:

$$\mathcal{L}_{\text{Judge-PG}} = -\mathbb{E}_{x\sim D}\mathbb{E}_{y\sim\pi_\theta}\left[\log\pi_\theta(y_{k_x}|x,y_{<k_x})\Big|_{y_{k_x}=y_{k_x}^*}\right]. \tag{2}$$

**Rejection Sampling for RL Generalization.** We observe that SFT loss computes the conditional probability under fixed prefixes to apply teacher forcing, while the Judge-PG loss approximates the marginal probability by aggregating over diverse prefix to maximize expected rewards. To address this exploration bottleneck, we leverage rejection sampling to enhance model generalization through diversified prefix generation. Our approach systematically generates and filters diverse response candidates based on quality metrics and reject the samples that do not match the ground truth label. Formally, for the $i_{th}$ instruction ($i \in \{1.., N\}$) in the dataset, we generate $M$ response samples that satisfy the ground truth label $y^{(i,*)}$ to approximate the policy gradient loss:

$$L_{\text{PG}} = -\frac{1}{NM}\sum_{i=1}^{N}\sum_{j=1}^{M}\log\pi_\theta(y_{k_x}^{(i,*)}|x^{(i)},y_{<k_x}^{(i,*)}) \tag{3}$$

We further combine the SFT loss and policy gradient loss as the total loss over the sampled responses:

$$\mathcal{L}_{total} = \mathcal{L}_{\text{SFT}} + \mathcal{L}_{\text{PG}} \quad . \tag{4}$$

**Mapping Function.** The total loss can be decomposed with SFT loss over the prefix and a mapping function $g$ over the prediction position, as shown in Table 1. We also design three different mapping loss function as $g$ over the ground truth answer $y_{k_{ij}}^{(i,*)}$ and the wrong answer $y_{k_{ij}}^{(i,-)}$.

• *DPO Mapping Loss w/o Reference Model* encourages the model to increase the probability of true answer while decreasing the probability of wrong answer.

- *Temperature Mapping Loss* performs temperature scaling to the logits before softmax, effectively sharpening the probability distribution around the ground truth token with $\tau$ as the temperature.

- *Margin Mapping Loss* introduces a margin $\gamma$ between the ground truth token and other answer, ensuring that the ground truth probability is sufficiently higher.

### 3.2.2 DATA PIPELINE

**Data Curation.** We begin by collecting open-source judge-related datasets, including Public Judge Data that contain critiques and Public Reward Data that only contain ground truth labels. For **Public Judge Data**, we observe that many judgments were generated by outdated models such as ChatGPT, which may introduce misjudgments and implicit errors. To address this issue, we split the data into outdated and up-to-date subsets based on the cutoff date of October 2024. For outdated data, we use Qwen2.5-72B-Instruct to reconstruct outdated judgment and verify correctness by comparing the predictions with human-labeled ground truth. For up-to-date data, we replace the original prompt templates with ones collected from a large number of subjective datasets available in the community, thereby enhancing their diversity. For **Public Reward Data**, such data lacks critique annotations, making it suboptimal for training generative judge models. To leverage this data effectively, we prompt Qwen2.5-72B-Instruct to generate judgments for each data instance. To further enhance the robustness and versatility, we systematically design and synthesize data from **Chat-based Datasets**, aiming to improve stylistic adaptability of the training data. Specifically, we generate response pairs exhibiting different styles and require Qwen2.5-72B-Instruct to select the superior response according to specified style requirements, thereby creating style-sensitive judge data.

**Rejection Sampling.** We apply the rejection sampling strategy mentioned above to ensure the accuracy of the reasoning paths. Specifically, for each question, we generate multiple judgments and filter out low-quality reasoning trajectories by checking whether the final predictions match the ground truth label provided in the original data.

### 3.3 LLM AS RUBRIC REFINER

#### 3.3.1 RUBRIC REFINEMENT TASK

An effective generalist judge model should not only be able to judge the responses, but also be capable of generating high-quality critiques that offer insightful analysis and explanations. For a given candidate model, it is crutial to evaluate the quality of feedback it provided in its role as a teacher during test time. For objective tasks, rule-based rewards are obtained in RLVR, which is not suitable for open-ended tasks. Therefore, we focus on the test-time improvement of student models for open-ended questions in our setting.

To make the refinement pipeline formalized and applicable, we transform the task into rubric generation. Specifically, we leverage rubrics generated by the candidate model to guide student models in refining their original responses. To quantitatively measure the improvement brought by the rubrics, we introduce a reference model and its responses as a point of comparison. We compute the win rate of the student model against the reference model both before and after refinement. The change in win rate reflects the effectiveness of the candidate model as a rubric refiner.

#### 3.3.2 DATA PIPELINE

**Data Curation.** We collect high-quality prompts from Condor (Cao et al., 2025) and utilize Qwen3-8B and Qwen3-30B-A3B to obtain student responses, which are then fed into the teacher model Qwen3-235B-Instruct (Yang et al., 2025) for rubric generation. In practice, however, we observe that LLMs tend to produce generic and superficial suggestions when only the question and response are provided, due to the lack of reference. Therefore, we adopt a **delta-based** strategy in our pipeline to improve the quality of the generated rubrics. Specifically, we first prompt Qwen3-235B-Instruct to generate an answer to the given question. Then, we ask the model to produce a rubric for potential improvement by comparing the student's response with its own generated answer. In this prompt, we require the model to avoid mentioning any reference to the self-generated answer, which facilitates the replacement of the prompt to obtain rubric-related training dialogue data based solely on the student's response.

**Rejection Sampling.** We adopt a rejection sampling strategy to filter for high-quality rubrics as well. Following the evaluation pipeline in RubricBench, we generate student responses alongside reference answers produced by DeepSeek-V3.1 at the initial stage. After the student responses are refined with rubric, we compare the improved responses against the original reference answers and select only those rubrics that successfully elevate the student response from being worse than to better than the reference answer. To prevent reward hacking, we use Qwen3-235B-Instruct itself as the judge, instead of GPT-OSS-120B as used in RubricBench.

### 3.4 OVERALL TRAINING DATA COMPOSITION.

Prior studies (Cao et al., 2024; Liu et al., 2025) have also demonstrated that incorporating general instruction data helps maintain a model's generalization capability while preserving its judge performance. Therefore, we also include general instruction data collected from CompassJudger-1 in our training dataset. The final training data for CompassJudger-2 consists of four components: (1) Objective verification data from knowledge-based datasets, (2) Subjective evaluation data from public judge and public reward datasets, (3) Rubric refinement data from delta-based strategy, (4) General supervised fine-tuning data.

### 3.5 MORE ROBUST EVALUATION

#### 3.5.1 JUDGERBENCHV2 FOR SUBJECTIVE EVALUATION

There are several limitations in existing benchmarks for judge models, such as insufficient coverage of judgment scenarios and noisy ground truth (GT) derived from a single source. To address these issues, we propose **JudgerBenchV2**, aiming to enhance the evaluation landscape for judge models by providing a more comprehensive and accurate benchmark. We first collect real-world user queries in both Chinese and English through CompassArena (Contributors, 2023). An LLM is then employed to classify each query by difficulty level, after which we manually select 100 queries per scenario, ensuring a balanced distribution of languages and difficulty levels. Next, we select 10 high-performing models with comparable capabilities and generate their responses to these queries. We then use GPT-4o-mini as the policy model and pair it with each of the 10 models to create response pairs. A judge model evaluates these pairs in a pairwise manner, and we derive its performance scores by comparing its decisions with GT.

Evaluating open-ended questions is inherently subjective, as different individuals and models may produce varying judgments. Relying solely on the judgment from a single human or a single model as GT therefore risks introducing bias. To address this issue, we introduce the **Mixture of Judgers** strategy, which leverages the judgments of DeepSeek-R1, DeepSeek-v3-0324, and Qwen3-235B-A22B and considers the majority consensus among these models as GT. Besides, traditional judge evaluation metrics primarily focus on sample-level accuracy and fail to capture essential dimensions such as ranking consistency. For instance, while human raters may disagree on individual samples, they often converge on the overall ranking of models. A comprehensive evaluation framework should therefore consider both fine-grained judgment accuracy and high-level ranking fidelity. We therefore calculate the score based on both accuracy and ranking. Please refer to Appendix for more details.

#### 3.5.2 RUBRICBENCH FOR RUBIRC REFINEMENT

Based on the settings mentioned above, we introduce **RubricBench**, a new benchmark for evaluating rubric refinement. We first collect high-quality real-world prompts from Arena-Hard, AlpacalEval, and CompassArena to construct our RubricBench dataset. We finally obtain 500 English questions and 500 Chinese queries. We then select Qwen3-8B, Qwen3-30B-A3B, Qwen3-4B-Instruct (Yang et al., 2025) and LLama3.1-8B (Grattafiori et al., 2024) as student models to generate corresponding responses and use Deepseek-V3.1 as the reference model for reference answers. We then employ GPT-OSS-120B as the judger to determine which answer is better and compute the win rate of the student models against the reference model.

For each given question and student response, the candidate model is required to generate a rubric consisting of 4 to 6 evaluation dimensions. Each dimension should be scored on a scale from 0 to 10, followed by a weighted overall score. In addition, the model should identify 1 to 2 weaker dimensions in the student's response, and provide targeted improvement suggestions for each of these

Table 2: **Main results on judge benchmarks.** CompassJudger-2 achieves state-of-the-art performance under different model scales across all the scenarios.

| Model | Objective Verification | Subjective Evaluation | | | | Rubric Refinement | Avg |
|---|---|---|---|---|---|---|---|
| | VerifyBench | JudgerBenchV2 | JudgeBench | RewardBench | Avg | RubricBench | |
| General LLMs | | | | | | | |
| Qwen2.5-7B-Instruct | 60.90 | 57.14 | 23.23 | 79.69 | 53.35 | 31.67 | 48.64 |
| Llama3.1-8B-Instruct | 46.90 | 57.64 | 33.23 | 73.64 | 54.84 | 39.18 | 46.97 |
| InternLM3-8B-Instruct | 58.40 | 57.71 | 24.19 | 80.62 | 54.84 | 40.74 | 51.33 |
| Qwen2.5-32B-Instruct | 72.00 | 62.97 | 59.84 | 85.61 | 69.47 | 45.42 | 62.30 |
| Verifier Models | | | | | | | |
| xVerify-0.5B-I | 77.90 | - | - | - | - | - | - |
| xVerify-8B-I | 83.20 | - | - | - | - | - | - |
| xVerify-9B-C | 83.20 | - | - | - | - | - | - |
| Tencent-Qwen2.5-7B-RLVR | 82.40 | - | - | - | - | - | - |
| Scalar Reward Models | | | | | | | |
| InternLM2-7B-Reward | - | 49.14 | 59.40 | 87.60 | 65.38 | - | - |
| InternLM2-20B-Reward | - | 45.89 | 63.40 | 90.20 | 66.50 | - | - |
| Skywork-Reward-Gemma-2-27B | - | 39.70 | 59.70 | 93.80 | 64.40 | - | - |
| Skywork-Reward-Llama-3.1-8B | - | 47.76 | 62.30 | 92.50 | 67.52 | - | - |
| 7B Judge Models | | | | | | | |
| CompassJudger-1-7B-Instruct | 60.40 | 57.96 | 46.00 | 80.74 | 61.57 | 44.58 | 55.52 |
| Con-J-7B-Instruct | 70.00 | 52.35 | 38.06 | 87.10 | 59.17 | 33.67 | 54.28 |
| Skywork-Critic-Llama-3.1-8B | 38.80 | 51.43 | 53.40 | 89.00 | 64.61 | 20.02 | 41.14 |
| RISE-Judge-Qwen2.5-7B | 61.70 | 46.12 | 40.48 | 88.20 | 58.27 | 41.11 | 53.69 |
| RM-R1-Qwen2.5-7B | 69.52 | 50.13 | 37.26 | 83.06 | 56.81 | 41.00 | 55.78 |
| CompassJudger-2-7B-Instruct | **85.50±1.04** | **61.25±0.32** | **61.13±0.45** | **89.14±0.39** | **70.51±0.39** | **46.03±0.43** | **67.35±0.62** |
| 32B+ Judge Models | | | | | | | |
| CompassJudger-1-32B-Instruct | 66.10 | 60.33 | 62.29 | 86.17 | 69.60 | 47.49 | 61.06 |
| Skywork-Critic-Llama-3.1-70B | 64.80 | 52.41 | 57.40 | 93.30 | 67.70 | 37.54 | 56.68 |
| RISE-Judge-Qwen2.5-32B | 72.00 | 56.42 | 63.87 | **92.70** | 70.99 | 44.91 | 62.63 |
| RM-R1-Qwen-Instruct-32B | 77.50 | 60.78 | 62.10 | 88.52 | 70.46 | 57.83 | 68.60 |
| CompassJudger-2-32B-Instruct | **85.60±0.87** | **63.45±0.28** | **64.84±0.34** | 91.35±0.28 | **73.21±0.30** | **63.00±0.51** | **73.94±0.54** |

Table 3: **Results on general benchmarks.** CompassJudger-2 maintains strong performance on both objective and subjective datasets.

| Model | MMLU Pro | GPQA Diamond | AIME25 | LiveCodeBench v5 | IFEval | ArenaHard | ChemBench | ProteinLM |
|---|---|---|---|---|---|---|---|---|
| 7B Judge Models | | | | | | | | |
| Qwen2.5-7B-Instruct | **55.43** | 34.85 | **6.67** | 12.57 | 73.20 | 47.86 | 57.20 | 57.52 |
| Con-J-7B-Instruct | 44.74 | 27.27 | 3.33 | 6.59 | 54.90 | 23.49 | 45.11 | **57.63** |
| RISE-Judge-Qwen2.5-7B | 51.56 | 32.32 | **6.67** | 12.57 | 44.18 | 35.99 | 57.72 | 54.24 |
| CompassJudger-2-7B-Instruct | 52.55 | **39.39** | **6.67** | **14.37** | **74.49** | **53.49** | **59.37** | 54.66 |
| 32B Judge Models | | | | | | | | |
| Qwen2.5-32B-Instruct | 68.92 | 42.93 | 16.67 | **30.54** | 79.85 | 70.16 | 71.75 | 61.23 |
| RISE-Judge-Qwen2.5-32B | 67.88 | 42.93 | 13.33 | 27.54 | 62.85 | 61.52 | **73.18** | 60.38 |
| CompassJudger-2-32B-Instruct | **69.22** | **50.51** | **23.33** | 25.15 | 79.48 | **83.31** | 72.42 | **61.65** |

identified areas. In this setting, the candidate model is expected to generate **rubrics with scoring logic** in a structured evaluation pipeline rather than merely listing scoring dimensions. We utilize models from different families as students, including Qwen, LLaMA, Gemma, and Kimi. Finally, we utilize the average of the relative improvement as the overall score of the candidate model. Please refer to Appendix for more details.

# 4 EXPERIMENTS

## 4.1 EXPERIMENTAL SETUP

**Evaluation Datasets.** We evaluate CompassJudger-2 on leading judge benchmarks, including VerifyBench (Yan et al., 2025), RewardBench (Lambert et al., 2024), JudgeBench (Tan et al., 2024), as well as our JudgerBenchV2 and RubricBench. Besides, we compare our method with other judge models over popular objective and subjective benchmarks, including MMLU Pro (Wang et al., 2024), GPQA (Rein et al., 2024), AIME2025, LiveCodeBench v5 (Jain et al., 2024), IFEval (Ding et al., 2023) and ArenaHard (Li et al., 2024).

**Experiment Settings.** In practice, we generate 8 candidate responses for filtering during rejection sampling. For model training, we utilize Qwen-2.5 series as the checkpoint and adopt 6e-5 as the learning rate. For Judge-PG loss, we set $\beta = 0.1$ in DPO Mapping Loss, $\tau = 5$ in Temperature Mapping Loss and $\gamma = 10$ in Margin Mapping Loss. We apply DPO Mapping Loss on only the candidate answer and Margin Mapping Loss on the top 10 logits. We train the model for 1 epoch

Table 4: **Ablation results of task data.** CompassJudger-2 benefits from different kinds of task data.

| Model | Objective Verification | Subjective Evaluation | | | | Rubric Refinement | Avg |
|---|---|---|---|---|---|---|---|
| | VerifyBench | JudgerBenchV2 | JudgeBench | RewardBench | Avg | RubricBench | |
| Objective Data Only | 86.60 | 57.52 | 52.74 | 49.94 | 53.40 | 34.13 | 58.04 |
| Subjective Data Only | 72.50 | 60.36 | 62.42 | 89.11 | 70.63 | 26.97 | 56.70 |
| Rubric Data Only | 69.60 | 57.75 | 50.16 | 44.25 | 50.72 | 42.00 | 54.11 |
| CompassJudger-2-7B-Instruct | 85.50 | 61.25 | 61.13 | 89.14 | 70.51 | 46.03 | 67.35 |

Table 5: **Different mapping loss.** Margin Mapping Loss performs better compared to others.

| Mapping Loss | JudgerBenchV2 | JudgeBench | RewardBench | Average |
|---|---|---|---|---|
| Baseline | 58.57 | 60.32 | 87.68 | 67.52 |
| DPO | 60.46 | 62.10 | 88.90 | 70.49 |
| Temperature | 60.93 | 61.61 | 88.58 | 70.37 |
| Margin | 61.25 | 61.13 | 89.14 | 70.51 |

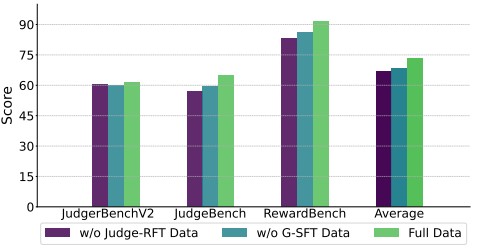

(a) Judge benchmarks.

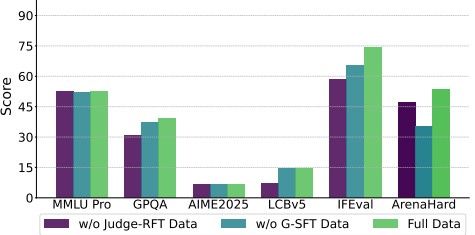

(b) Subjective and objective benchmarks.

Figure 3: **Data ablation results on different datasets.** Judge-RFT data lead to improvements on general benchmarks, whereas G-SFT data have little impact on judging capability.

with batch size equal to 256 with 64 GPUs. During evaluation, we set the temperature to 0.6 and perform three runs using different random seeds. We report the average performance with error bar.

## 4.2 MAIN RESULTS

**Judge Ability Analysis.** To verify the judge ability of our method, we conduct evaluation across multiple benchmarks and compare our method with general models and specialized judge models including the RM-R1 (Chen et al., 2025b) and RISE (Yu et al., 2025) series. As presented in Table 2, CompassJudger-2 consistently surpasses all baselines, demonstrating significant advancements in the generalization ability. Notably, CompassJudger-2-7B-Instruct outperforms RISE-Judge-Qwen2.5-Instruct-7B and RM-R1-Qwen-Instruct-7B by 15.48% and 13.39% on average, respectively.

**General Ability Analysis.** We further highlight the improvements in general capabilities of CompassJudger-2 compared to baselines across objective and subjective benchmarks. Besides, we evaluate the performance in several unseen domains including ChemBench (Mirza et al., 2025) and ProteinLM (Shen, 2024).As shown in Table 3, CompassJudger-2 achieves markedly superior performance over other judge models on both objective and subjective datasets, demonstrating its generalization ability. Remarkably, CompassJudger-2 surpasses general models like Qwen2.5-Instruct-7B-Instruct and Qwen2.5-32B-Instruct on specific datasets, revealing a strong correlation between judge ability and general ability in LLMs and their potential to enhance each other.

## 4.3 ABLATION AND DISCUSSION

**Task Ablation.** To better understand how different task data components contribute to the performance, we conduct additional ablation analysis and report the results in Table 4. The results show that different types of data lead to noticeable improvements on their corresponding tasks. Notably, although the all-in-one model achieves performance comparable to ablation variants on objective verification and subjective evaluation, it significantly outperforms them on RubricBench.

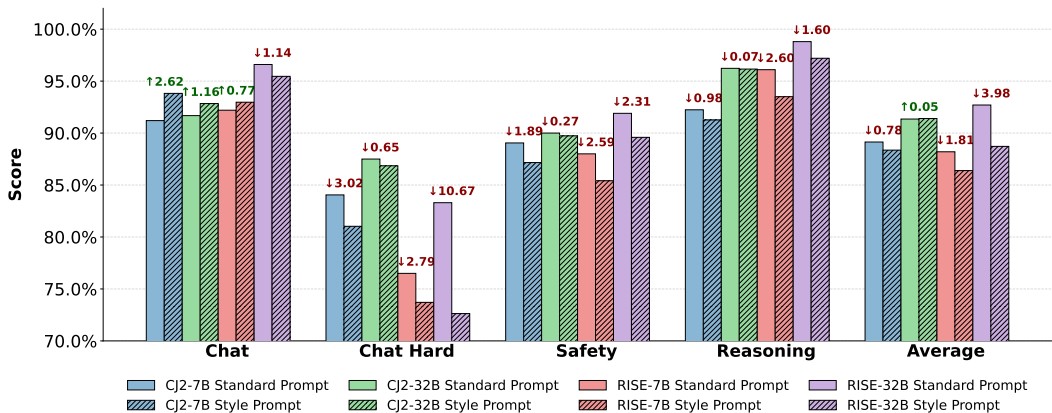

Figure 4: **Comparison of style judge performance between CompassJudger-2 and RISE.** CompassJudger-2 demonstrates greater robustness than RISE across different judging prompts.

Table 6: **Results on RubricBench.** Qwen3-4B-IT, Gemma3-4b-IT and Kimi-Linear denotes Qwen3-4B-Instruct, Gemma3-4b-Instruct and Kimi-Linear-48B-A3B-Instruct, respectively. CompassJudger-2 outperforms baselines by a substantial margin.

| Models | Qwen3-8B | Qwen3-30B-A3B | Llama3.1-8B | Qwen3-4B-IT | Gemma3-4B-IT | Kimi-Linear | Overall |
|---|---|---|---|---|---|---|---|
| **Base Models** | | | | | | | |
| Base model | 16.70 | 15.90 | 8.10 | 25.90 | 16.00 | 30.80 | 0.00 |
| **7B–8B Models** | | | | | | | |
| Qwen2.5-Instruct-7B | 25.00 | 23.10 | 6.35 | 39.70 | 18.80 | 36.60 | 27.17 |
| Con-J-7B-Instruct | 23.10 | 25.50 | 6.70 | 39.70 | 19.50 | 37.20 | 29.56 |
| LLama3.1-8B | 24.60 | 26.80 | 6.50 | 41.60 | 19.90 | 40.10 | 35.22 |
| InternLM3-8B-Instruct | 24.10 | 23.90 | 8.10 | **43.60** | 21.00 | 40.00 | 37.35 |
| CompassJudger-1-7B-Instruct | 24.20 | 26.70 | 8.15 | 42.70 | 20.70 | 39.10 | 39.11 |
| Skywork-Critic-Llama-3.1-8B | 23.70 | 21.50 | 7.80 | 36.80 | 14.40 | 35.30 | 20.02 |
| RM-R1-Qwen2.5-Instruct-7B | 26.00 | 26.60 | 6.20 | 42.60 | 18.60 | 40.20 | 35.13 |
| RISE-Judge-Qwen2.5-Instruct-7B | 27.20 | 24.80 | 6.60 | 42.50 | 20.40 | 38.80 | 36.32 |
| **CompassJudger-2-7B-Instruct** | **27.40 ± 0.50** | **27.30 ± 0.45** | **9.00 ± 0.30** | 41.20 ± 0.25 | **22.20 ± 0.40** | **40.50 ± 0.35** | **46.03 ± 0.43** |
| **32B Models** | | | | | | | |
| Qwen2.5-Instruct-32B | 27.00 | 27.00 | 7.80 | 42.00 | 21.50 | 39.40 | 42.04 |
| CompassJudger-1-32B-Instruct | 25.40 | 26.20 | 8.00 | 43.00 | 20.30 | 39.00 | 39.19 |
| Skywork-Critic-Llama-3.1-70B | 23.90 | 26.20 | 8.10 | 43.40 | 19.50 | 39.40 | 37.54 |
| RM-R1-Qwen2.5-32B | 27.50 | 27.30 | 9.60 | 45.70 | 23.00 | 41.60 | 51.69 |
| RISE-Judge-Qwen2.5-32B | 25.60 | 26.20 | 7.80 | 42.80 | 21.80 | 39.60 | 40.74 |
| **CompassJudger-2-32B-Instruct** | **29.70 ± 0.45** | **28.80 ± 0.48** | **11.50 ± 0.50** | **47.50 ± 0.51** | **24.80 ± 0.55** | **42.70 ± 0.57** | **63.00 ± 0.51** |

This suggests that the rubric refinement task builds upon both objective verification and subjective evaluation, and that all three types of task data jointly contribute to the improvement.

**Judge-PG Loss.** To evaluate the impact of incorporating policy gradient loss, we conduct a detailed ablation study to identify the most effective type of mapping loss for improving model performance, as shown in Table 5. Our findings indicate that the verified reward serves as a crucial supervised signal, significantly enhancing the model's performance on judgment tasks. All models trained with policy gradient loss outperform the baseline on RewardBench, achieving performance gains of up to 1.5%. Notably, the model trained with margin loss demonstrates slightly better generalization, yielding an average performance improvement of 2.99% over the baseline. Consequently, we adopt margin loss as the default choice in our study.

**General Ability Benefits from Judge Data.** To investigate how general supervised fine-tuning data (G-SFT Data) and rejection sampling judge data (Judge-RFT Data) impact judge ability and general ability, we perform ablation studies by separately removing each data type from the training set. As illustrated in Figure 3, the results highlight several key findings. Removing RFT data causes a significant decline in judge performance. In addition, including RFT data enhances performance across specific datasets, such as GPQA-Diamond and ArenaHard, underscoring its role in boosting general ability. In contrast, General SFT data primarily maintain the general ability of the model, with minimal impact on judge ability.

**Better Robustness with Style Judging Prompts.** An effective generalist judge model should also maintain consistent performance with various prompts. Therefore, we conduct style judge experiment of different prompts by adding following sentences: "Beyond this, users prefer a more detailed response; therefore, you need to determine which model's answer provides more comprehensive and useful information when both responses are correct and have completed the user's request". We present the results on different subset of RewardBench. As can be seen from the results in Figure 4, RISE-Judge-Qwen2.5-32B suffers from significant performance drop by 10.67% in the Chat Hard subset. Compared with RISE, CompassJudger-2 are less sensitive of judging prompts and show better consistency and generalization ability, indicating the superiority of our method.

**Generalization to OOD Rubric Refinement.** In Table 6, we present the performance improvements achieved by various student models after rubric refinement. The overall score is calculated as the average relative improvement across all student models. The performance of Qwen3-8B and Qwen3-30B-A3B reflects in-distribution behavior, as these models are involved in both training data construction and evaluation. In contrast, the performance of Qwen3-4B-Instruct, LLaMA3.1-8B, Gemma3-4B-IT, and Kimi-Linear-48B-A3B-Instruct, which are not included in the training data construction, demonstrates CompassJudger-2's ability to generalize to out-of-distribution scenarios. As shown in the results, CompassJudger-2's rubric refinement capability is robust and generalizes well to student models it has never seen before, demonstrating our superior performance.

## 5 CONCLUSIONS

In this work, we present CompassJudger-2, an series of all-in-one judge models that advance LLM-as-judge performance through a unified training paradigm combining diverse task-driven data composition, high-quality chain-of-thought supervision, and verifiable reward-guided optimization. Furthermore, we introduce JudgerBenchV2, a comprehensive benchmark with mixed-of-judgers and novel ranking-aware metrics, to enable more nuanced and reliable evaluation of judge models. Looking forward, CompassJudger-2 paves the way for more adaptable, interpretable, and efficient judge services in real-world LLM deployments, and we anticipate that extending this work to multi-modal and interactive evaluation scenarios will further enhance its applicability and impact.

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

# APPENDIX

## A  LLM USAGE

In the preparation of this manuscript, we made limited use of LLMs as a general-purpose writing assistant. Specifically, the LLMs are employed to polish wording, improve sentence fluency, and adjust grammatical structure for clarity and readability. At no point did the LLMs contribute to research ideation, the formulation of hypotheses, methodological design, execution of experiments, or interpretation of results. Their role was strictly confined to surface-level language refinement, comparable to the functions of a grammar-checking or style-editing tool. All intellectual contributions, including the conception of ideas, development of approaches, and analysis of findings, are entirely the work of the authors.

## B  ETHICS STATEMENT

In this work, we introduce two benchmarks, namely JudgerBenchV2 and RubricBench. The data used for evaluation have been carefully curated, and we confirm that they contain no harmful, offensive, or personally identifiable information. All collection procedures respect community guidelines and prevailing ethical standards in AI research.

In particular, we obtained the datasets from publicly available, non-sensitive sources. No private user conversations, restricted-access materials, or proprietary data are included. Any data filtering was performed with the explicit aim of reducing potential biases, mitigating harmful content, and ensuring that the resulting benchmarks provide fair and transparent evaluation. We emphasize that these benchmarks are intended strictly for research purposes. They are designed to facilitate the study of evaluation methods and model performance, not to promote or enable misuse. The benchmarks should not be repurposed for applications that could negatively affect individuals, groups, or societal well-being. Finally, we align this work with the broader principles of responsible AI research: transparency, reproducibility, and accountability. Wherever applicable, we make our data construction process clear and document potential limitations or biases so that future researchers can build upon our work in a safe and ethically sound manner.

## C  EVALUATION METRIC

### C.1  JUDGERBENCHV2

In JudgerBenchV2, we conduct pairwise comparisons between a candidate model and GPT-4o-mini to determine which delivers superior responses. Each comparison is evaluated by both a ground truth judge model and a test judge model. A sample is considered correct if both judges agree on the better-performing model. For each sample, the model deemed superior earns a score increment of 1. The total number of pairwise samples is denoted by $N$ and $C$ represents the number of samples where the GT and test judge models agree on the superior model. For a set of $M$ candidate models, let the GT judge model and the test judge model generate score lists $S_1 = \{s_{1,m}\}_{m \in M}$ and $S_2 = \{s_{2,m}\}_{m \in M}$, respectively, where $s_{i,m}$ represents the cumulative score for model $m$ based on pairwise wins. Additionally, let $R_1 = \{r_{1,m}\}_{m \in M}$ and $R_2 = \{r_{2,m}\}_{m \in M}$ denote the rank lists, where $r_{i,m}$ is the rank of model $m$ according to judge $i$. The performance of the test judge model is evaluated using the following metric:

$$
\mathcal{P} = 100 \cdot \underbrace{\frac{C}{N}}_{\substack{\text{Sample-level} \\ \text{accuracy}}} - \frac{100}{|M|} \sum_{m \in M} (* \underbrace{\frac{|r_{1,m} - r_{2,m}|}{|M| - 1}}_{\substack{\text{Normalized rank} \\ \text{difference}}} + \underbrace{\frac{|s_{1,m} - s_{2,m}|}{\max_{m' \in M} |s_{1,m'} - s_{2,m'}|}}_{\text{Normalized score difference}}).
\tag{5}
$$

The first term captures the sample-level accuracy by measuring the agreements between the judges. The second term penalizes discrepancies in rankings and scores, with normalization to ensure equitable comparisons across different models.

## C.2 RUBRICBENCH

In RubricBench, we use the average relative improvement compared with the Base model as the metric to calculate the overall score of the teacher model. It measures how much, on average, a given teacher model improves over the baseline in relative percentage.

Let $s_{i,j}$ to be score of teacher model $i$ under student model $j$ and $s_{\text{base},j}$ to be score of the base student model $j$. Then the relative improvement for student model $j$ is defined as:

$$r_{i,j} = \frac{s_{i,j} - s_{\text{base},j}}{s_{\text{base},j}} \times 100\%.$$

We can then obtain the overall score for model $i$ is the mean of these improvements across all $N$ student models:

$$\text{Overall}_i = \frac{1}{N} \sum_{j=1}^{N} r_{i,j}.$$

# D DERIVING THE LOSS FUNCTION

**Judge Reward.** In the judge task, given a instruction-response pair $(x, y)$, prediction position $k_x$ and ground truth label $y_k^*$, we apply a rule-based reward defined as:

$$r(x, y) = \begin{cases} 1 & \text{if } y_{k_x} = y_{k_x}^* \\ 0 & \text{otherwise} \end{cases}. \tag{6}$$

**Policy Gradient Optimization.** To optimize the judge model's performance, we formulate the learning objective as maximizing the expected reward over the response distribution:

$$J(\theta) = \mathbb{E}_{x \sim D} \mathbb{E}_{y \sim \pi_\theta} [r(x, y)] \tag{7}$$

The gradient of this objective can be derived using the policy gradient theorem:

$$\nabla_\theta J(\theta) = \mathbb{E}_{x \sim D} \mathbb{E}_{y \sim \pi_\theta} [r(x, y) \nabla_\theta \log \pi_\theta(y|x)]$$
$$= \mathbb{E}_{x \sim D} \mathbb{E}_{y \sim \pi_\theta} \left[ r(x, y) \sum_{t=1}^{n} \nabla_\theta \log \pi_\theta(y_t|x, y_{<t}) \right] \tag{8}$$

This decomposition shows how the gradient propagates through all sequence positions in autoregressive models. The corresponding policy gradient loss is:

$$\mathcal{L}_{\text{PG}} = -\mathbb{E}_{x \sim D} \mathbb{E}_{y \sim \pi_\theta} \left[ r(x, y) \sum_{t=1}^{n} \log \pi_\theta(y_t|x, y_{<t}) \right] \tag{9}$$

Given our binary reward function that only depends on the prediction at position $k_x$, we can simplify:

$$\mathcal{L}_{\text{PG}} = -\mathbb{E}_{x \sim D} \mathbb{E}_{y \sim \pi_\theta} [r(x, y) \log \pi_\theta(y_{k_x}|x, y_{<k_x})]$$
$$= -\mathbb{E}_{x \sim D} \mathbb{E}_{y \sim \pi_\theta} [\mathbb{I}(y_{k_x} = y_{k_x}^*) \log \pi_\theta(y_{k_x}|x, y_{<k_x})]$$
$$= -\mathbb{E}_{x \sim D} \mathbb{E}_{y \sim \pi_\theta} \left[ \log \pi_\theta(y_{k_x}|x, y_{<k_x}) \Big|_{y_{k_x} = y_{k_x}^*} \right] \tag{10}$$

**Rejection Sampling for RL Generalization.** We further apply rejection sampling to approximate the policy gradient loss. Formally, for the $i_{th}$ instruction ($i \in \{1.., N\}$) in the dataset, we generate $M$ response samples that satisfy the ground truth label $y^{(i,*)}$ and obtain the following loss:

$$L_{\text{PG}} = -\mathbb{E}_{x \sim D} \mathbb{E}_{y \sim \pi_\theta} \left[ \log \pi_\theta(y_{k_x}|x, y_{<k_x}) \Big|_{y_{k_x} = y_{k_x}^*} \right]$$
$$= -\frac{1}{N} \sum_{i=1}^{N} \frac{1}{M} \sum_{j=1}^{M} \left[ \log \pi_\theta(y_{k_x}^{(i,j)}|x^{(i)}, y_{<k_x}^{(i,j)}) \Big|_{y_{k_x}^{(i,j)} = y_{k_x}^{(i,*)}} \right] \tag{11}$$
$$= -\frac{1}{NM} \sum_{i=1}^{N} \sum_{j=1}^{M} \log \pi_\theta(y_{k_x}^{(i,*)}|x^{(i)}, y_{<k_x}^{(i,*)})$$

Similarly, we apply the SFT loss to the sampled response candidates. To balance the standard sequence modeling objective with reward optimization, we combine the SFT loss and policy gradient loss through a mapping function $f$ and derive another mapping function $g$:

$$
\begin{aligned}
\mathcal{L} &= \mathcal{L}_{\text{SFT}} + \mathcal{L}_{\text{PG}}(f) \\
&= -\frac{1}{NM} \sum_{i=1}^{N} \sum_{j=1}^{M} \sum_{t \neq k_{ij}} \log \pi_\theta(y_t^{(i,j)}|x^{(j)}, y_{<t}^{(i,j)}) \\
&\quad - \frac{1}{NM} \sum_{i=1}^{N} \sum_{j=1}^{M} \log \pi_\theta(y_{k_{ij}}^{(i,*)}|x^{(j)}, y_{<k_{ij}}^{(i,j)}) + f\left(\log \pi_\theta(y_{k_{ij}}^{(i,*)}|x^{(j)}, y_{<k_{ij}}^{(i,j)})\right) \\
&= -\frac{1}{NM} \sum_{i=1}^{N} \sum_{j=1}^{M} \left[ \sum_{t \neq k_{ij}} \log \pi_\theta(y_t^{(i,j)}|x^{(j)}, y_{<t}^{(i,j)}) + g\left(\log \pi_\theta(y_{k_{ij}}^{(i,*)}|x^{(j)}, y_{<k_{ij}}^{(i,j)})\right) \right]
\end{aligned}
\tag{12}
$$

where $g$ is a composite function that combines the original mapping function $f$ with the log probability term to provide a more flexible optimization objective. In our method, the mapping function $g$ is approximate by DPO loss, Temperture Loss and Margin Loss.

## E DETAILED RESULTS ON THE JUDGE BENCHMARKS

We list the detailed results of judge models on the Judge Benchmarks.

Table 7: **Detailed results on JudgerBenchV2 benchmarks.**

| Model | Accuracy | Normalized Diff | Rank Diff | Score Diff | Final Score |
|---|---|---|---|---|---|
| **7B Judge Models** | | | | | |
| CompassJudger-1-7B-Instruct (Cao et al., 2024) | 77.41 | 61.48 | 11.40 | 83.40 | 57.96 |
| Skywork-Critic-Llama-3.1-8B (Shiwen et al., 2024) | 66.68 | 63.82 | 16.20 | 112.70 | 51.43 |
| Con-J-Qwen2-7B (Ye et al., 2025) | 71.30 | 66.61 | 17.60 | 85.20 | 52.35 |
| RISE-Judge-Qwen2.5-7B (Yu et al., 2025) | 70.08 | 77.85 | 14.00 | 202.50 | 46.12 |
| CompassJudger-2-7B-Instruct | $78.04 \pm 0.21$ | $57.00 \pm 0.80$ | $10.80 \pm 0.02$ | $76.90 \pm 4.22$ | $60.52 \pm 0.32$ |
| **32B+ Judge Models** | | | | | |
| CompassJudger-1-32B-Instruct (Cao et al., 2024) | 80.99 | 60.32 | 11.40 | 62.90 | 60.33 |
| Skywork-Critic-Llama-3.1-70B (Shiwen et al., 2024) | 70.27 | 65.44 | 15.20 | 97.30 | 52.41 |
| RISE-Judge-Qwen2.5-32B (Yu et al., 2025) | 74.00 | 61.15 | 10.60 | 88.80 | 54.42 |
| CompassJudger-2-32B-Instruct | $80.90 \pm 0.23$ | $56.47 \pm 0.52$ | $8.60 \pm 0.07$ | $64.10 \pm 3.15$ | $62.21 \pm 0.28$ |

Table 8: **Detailed results on RewardBench benchmarks.**

| Model | Chat | Chat Hard | Safety | Reasoning | Final Score |
|---|---|---|---|---|---|
| **7B Judge Models** | | | | | |
| CompassJudger-1-7B-Instruct (Cao et al., 2024) | 97.80 | 61.00 | 84.50 | 89.50 | 83.20 |
| Skywork-Critic-Llama-3.1-8B (Shiwen et al., 2024) | 93.60 | 81.40 | 91.10 | 89.80 | 89.00 |
| Con-J-Qwen2-7B (Ye et al., 2025) | 91.90 | 80.30 | 88.20 | 88.10 | 87.10 |
| RISE-Judge-Qwen2.5-7B (Yu et al., 2025) | 92.20 | 76.50 | 88.00 | 96.10 | 88.20 |
| CompassJudger-2-7B-Instruct | $91.20 \pm 0.33$ | $84.05 \pm 0.57$ | $89.05 \pm 0.21$ | $92.24 \pm 0.39$ | $89.14 \pm 0.28$ |
| **32B+ Judge Models** | | | | | |
| CompassJudger-1-32B-Instruct (Cao et al., 2024) | 98.00 | 65.10 | 85.30 | 92.40 | 85.20 |
| Skywork-Critic-Llama-3.1-70B (Shiwen et al., 2024) | 96.60 | 87.90 | 93.10 | 95.50 | 93.30 |
| RISE-Judge-Qwen2.5-32B (Yu et al., 2025) | 96.60 | 83.30 | 91.90 | 98.80 | 92.70 |
| CompassJudger-2-32B-Instruct | $91.68 \pm 0.23$ | $87.50 \pm 0.55$ | $90.00 \pm 0.28$ | $96.23 \pm 0.41$ | $91.35 \pm 0.39$ |

**CoT Synthesizing Prompt**

Now we are reviewing a user's interaction with two models. Your task is to evaluate the responses from Model A and Model B by carefully analyzing the dialogue step by step, following a clear and structured thought process:
1. User's Demand:
- Carefully analyze the user's request. What is the user specifically asking for? What are the key aspects of the request that need to be fulfilled? Identify any constraints (e.g., time, format, quantity) the user has provided.
2. Strengths of Model A:
- Identify the strengths of Model A's response. Consider how well it addresses the user's demand, meets the user's constraints, and how well it serves the intended purpose.
3. Weaknesses of Model A:
- Identify the weaknesses of Model A's response. What aspects of the response fail to meet the user's request or constraints? What could have been improved?
4. Strengths of Model B:
- Identify the strengths of Model B's response. Consider how well it addresses the user's demand, meets the user's constraints, and how well it serves the intended purpose.
5. Weaknesses of Model B:
- Identify the weaknesses of Model B's response. What aspects of the response fail to meet the user's request or constraints? What could have been improved?
6. Reasoning:
- Based on your analysis of both responses, explain which model better addresses the user's needs. Discuss which model's response is more suitable given the user's request and constraints.
7. Choice:
- Conclude with a choice between Model A and Model B based on your reasoning. Indicate which model provides the more appropriate and useful response for the user's request.
Your final reply must be structured in the following format:
{
"User's Demand": "[The user's request or need]",
"Strengths of Model A": "[Summary of the strengths of Model A]",
"Weaknesses of Model A": "[Summary of the weaknesses of Model A]",
"Strengths of Model B": "[Summary of the strengths of Model B]",
"Weaknesses of Model B": "[Summary of the weaknesses of Model B]",
"Reasoning": "[Explanation of which model is more suitable for the user's demand]",
"Choice": "[Model A or Model B]"
}

---

**Delta-based Rubric Generation Prompt**

[[ Instruction ]]
You will receive the following three inputs: a user question, a model-generated response, and a reference answer (serving as an implicit quality benchmark). Your core tasks are:
1. **Generate a customized scoring rubric by rigorously comparing the model response to the reference answer** (all dimensions must be derived exclusively from the user question, reflecting its core requirements; the reference answer is only used internally to identify high-quality features, but must not be mentioned or implied in the rubric).
2. **Use this rubric to objectively evaluate the quality of the given model response**. The evaluation must be based solely on the model response and the user's question. **You must not reveal the existence of the reference answer or imply any comparison** (e.g., avoid phrases like "should include" or "correct approach").
3. **Provide specific, actionable improvement suggestions** based on the evaluation, directly guiding how to enhance the response.
Strict rules for output:
- **Absolutely no reference to the reference answer**: Avoid terms like "reference answer", "example", "benchmark", "high-quality", "should include", or any implicit comparison. All conclusions must stem solely from the analysis of the **given model response** against the scoring rubric, as if you only received the user question and model response (i.e., pretend the reference answer does not exist).
- **Scoring rubric must be deeply customized to the question**: Dimensions must closely align with the question type and be directly derived from it. Avoid generic templates. The reference answer may only be used as an internal aid and must not be described or referenced in the rubric.
- **Scoring must identify real weaknesses**: In the step-by-step scoring, **not all dimensions may receive a perfect 10**. At least one dimension must score <=8 to demonstrate room for improvement. Even for high-quality responses, potential optimization points must be identified.
- The scoring rubric must be presented in a **table format**, containing **4–6 specific dimensions**. Each dimension must include:
- **Description** (how it serves the question's needs)
- **Scoring criteria** (0–10 scale with clear judgment benchmarks)
- **Weight** (summing to 100%, reasonably allocated based on question priorities)
Final score = weighted average of dimension scores.
- Output must contain exactly three parts:
a. **Scoring Rubric Table**: Fully list dimensions, descriptions, criteria, and weights.
b. **Step-by-Step Scoring Process**: For each dimension, explain the score based on the given response, **citing exact text from the response**. **Explicitly state at least one dimension scoring <= 8 and justify why**.
c. **Final Score and Improvement Suggestions**:
- Provide total score and brief summary.
- **Mandatory Weakness Identification**: List all dimensions with scores <=8 (at least one), and explain specific weaknesses using quotes from the response.
- **Targeted Improvement Suggestions**: For each weak dimension, provide **1–2 action-able suggestions** that are:
- **Specific to text location** (e.g., "in the section '...'")
- **Executable** (clear steps to fix)
- **Not generic** (avoid ineffective phrases like "could be more detailed" or "suggest improving logic")
[[ User Question Begin ]]
question
[[ User Question End ]]
[[ Model Response Begin ]]
response
[[ Model Response End ]]
[[ Reference Answer Begin ]]
reference
[[ Reference Answer End ]]

---

---

**RubricBench Evaluation Prompt**

[[ Instruction ]]
You will receive the following two parts of input: a question posed by the user, and a model-generated response. Your task is to:
1. **Generate a customized scoring rubric based on the user's question** (dimensions must be derived from the question and reflect its core requirements).
2. **Use this rubric to objectively evaluate the quality of the given response**.
3. **Provide specific, actionable improvement suggestions** based on the evaluation, directly guiding how to enhance the response.
In your output, strictly adhere to the following:
- **Do not mention any other models, comparative content, or external references** (e.g., avoid phrases like "better than" or "correct answer"). All conclusions must be drawn solely from the analysis of the **given response** using the scoring rubric.
- **The scoring rubric must be customized to the question**: dimensions must be closely tied to the question type; avoid generic templates.
- **The evaluation must identify real weaknesses**: in the step-by-step scoring, **not all dimensions may score a perfect 10**; at least one dimension must score <= 8 to reflect room for improvement. Even if the response is high-quality, potential optimization points must be identified.
- The scoring rubric must be presented in **table format**, containing **4–6 specific dimensions**. Each dimension must include:
- **Description** (how the dimension serves the needs of the question)
- **Scoring criteria** (on a 0–10 scale, with clear judgment guidelines for each score)
- **Weight** (summing to 100%, reasonably allocated to reflect the priority implied by the question)
Final score = weighted average of dimension scores × weights.
- Your output must contain the following three sections:
a. **Scoring Rubric Table**: fully list dimensions, descriptions, scoring criteria, and weights.
b. **Step-by-Step Scoring Process**: for the given response, explain the score for each dimension **with direct quotes from the response**, and calculate the score per dimension. **Explicitly identify at least one dimension scoring <= 8 and explain why**.
c. **Final Score and Improvement Suggestions**:
- Provide the final total score and a brief summary.
- **Mandatory Weakness Identification**: list all dimensions with scores <=8 (at least one), and explain the specific weaknesses using excerpts from the response.
- **Targeted Improvement Suggestions**: for each weak dimension, provide **1–2 actionable suggestions** that are:
- **Specific to the location in the response text**
- **Executable**
- **Not generic** (avoid ineffective phrases like "could be more detailed" or "suggest improving logic")
[[ User Question Begin ]]
question
[[ User Question End ]]
[[ Model Response Begin ]]
response
[[ Model Response End ]]

