# OpenReview forum: "CompassJudger-2: A Holistic Approach Towards Generalist Judge Model"
_ICLR.cc/2026/Conference — Submitted to ICLR 2026_

### Official Review · Reviewer_6U2P · 2025-10-18

**Soundness:** 3
**Presentation:** 2
**Contribution:** 2
**Rating:** 4
**Confidence:** 4

**Summary:**

This work introduces CompassJudger-2, a new generalist judge model, alongside a comprehensive data curation strategy and two new benchmarks (JudgerBenchV2 and RubricBench). The authors decompose the capabilities of a judge model into three distinct scenarios: objective verification, subjective evaluation, and rubric refinement, and develop a tailored data collection and filtering pipeline for each. The resulting model is evaluated on a suite of existing and newly proposed benchmarks to demonstrate its robustness and generalization.

**Strengths:**

The paper presents a comprehensive investigation into the LLM-as-a-Judge paradigm. By systematically decomposing judge capabilities and addressing a wide range of evaluation scenarios (objective, subjective, and rubric-based), the work provides broad and valuable coverage of the problem space.

**Weaknesses:**

1. As the paper's core contributions include a new dataset and two new benchmarks, their availability is crucial for verification and for the broader research community. It is highly recommended that the authors make these resources publicly available, at least during the rebuttal phase, to allow reviewers to fully assess their quality and impact.

2. The manuscript could be improved in terms of clarity and structure. For instance, the transition and logical connection between Sections 3.2.1 ("Incorporating Verified Reward") and 3.2.2 ("JudgerBenchV2") are not immediately clear, which may cause confusion for the reader. Strengthening the narrative bridges between different methodological sections would enhance readability.

**Questions:**

1. In Line 260, you state that rejection sampling for the **subjective evaluator** is performed by matching predictions with the ground truth. Could you elaborate on this process? Given that subjective questions are open-ended and may not have a single "correct" answer, how is this ground-truth matching implemented, and how robust is this strategy to the inherent ambiguity of subjective data?

2. The paper presents an ablation study on the entire Judge-RFT dataset. It would be more impactful to see a more fine-grained ablation that isolates the contribution of the specific data components mentioned in Section 3.4 (objective, subjective, rubric, and general SFT data). For example, how does training on each data component individually affect performance on the corresponding benchmarks (e.g., RewardBench and RubricBench)? Such an analysis would be very helpful in demonstrating the effectiveness of your task-specific data curation strategy.

3. Generalization to State-of-the-Art Models: It is noted that several state-of-the-art LLMs (like GPT-OSS and Qwen3) were used during the dataset creation phase. However, the experiments in the paper are conducted primarily on what might be considered legacy models (e.g., Qwen2.5, Llama-3.1). To convincingly demonstrate the generalization ability of your proposed datasets and benchmarks, it is highly recommended to include experimental results on some of the latest publicly available LLMs.

I would be happy to raise my score if the authors can address my concerns.

---

> ### Author Response · Authors · 2025-11-26
>
> **Q1. Data release.**
>
> Please refer to Q3 in Global Response for more details.
>
> **Q2. Clarity and structure.**
>
> Thanks for your advice. We have separated the construction of JudgerBenchV2 and RubricBench into an independent section. Section 3.2, which focuses on subjective evaluation, will now only include our methodology that incorporates verified reward and the data pipeline. The structure of the manuscript would be clearer and more readable through these revisions.
>
> **Q3. Explanation on subjective evaluation.**
>
> The training prompts and labels we use for subjective evaluation are mainly from public datasets. Their ground truth is typically obtained either through human annotation followed by a voting pipeline, or by introducing subtle factual errors into one of the responses in an A/B prompt pair. In fact, many benchmarks for subjective evaluation are constructed in a similar way.
>
> Therefore, we can consider the ground truth obtained through these methods to be largely aligned with human preferences.
>
> **Q4: Ablation on different task data.**
>
> Please refer to Q2 in Global Response for more details.
>
> **Q5: More student models in RubricBench.**
>
> Please refer to Q1 in Global Response for more details.

---

### Official Review · Reviewer_9wnn · 2025-10-28

**Soundness:** 3
**Presentation:** 3
**Contribution:** 3
**Rating:** 6
**Confidence:** 5

**Summary:**

This paper proposes CompassJudger-2, an all-in-one LLM-as-Judge framework designed to generalize across three skill hierarchies: objective verification, subjective evaluation, and rubric refinement. The authors introduce two new benchmarks—JudgerBenchV2 (for accuracy + rank-fidelity evaluation with “Mixture-of-Judgers” ground truth) and RubricBench (for test-time improvement through rubric refinement). The work builds upon CompassJudger-1 by expanding data diversity via task-specific rejection sampling and verifiable reward guidance. Experiments show state-of-the-art results on multiple judge and reward benchmarks.

**Strengths:**

1: Clear task hierarchy. The decomposition into objective verification, subjective evaluation, and rubric refinement is well motivated and pedagogically intuitive.

2: Contribution in New benchmarks. JudgerBenchV2 and RubricBench contribute practical evaluation infrastructure and metrics that better capture ranking consistency and rubric-guided improvement.

3: Moderate innovation at model + data levels. The model integrates verified rewards and delta-based rubric generation, and the datasets are carefully curated with rejection sampling for reliability.

4: Strong empirical results. The system achieves solid gains across most judge and reward benchmarks and maintains good robustness across different judging styles.

5: Well-structured related-work section. The literature review is clean and logically organized, situating the work clearly in the LLM-as-Judge landscape.

**Weaknesses:**

1: Unsubstantiated claim of domain generalization. The introduction argues that existing judge models “fail on unfamiliar tasks or domains” and that CompassJudger-2 remedies this, yet all experiments are performed on common and popular benchmarks (MMLU-Pro, GPQA, RewardBench, ArenaHard, etc.). To validate generalization, the model should be tested on genuinely out-of-domain or unseen domains (e.g., non-academic reasoning, multimodal judgment, low-resource languages).

2: Pedagogical analogy lacks theoretical grounding. The paper draws inspiration from “the hierarchy of skills demonstrated by teachers,” but no pedagogical or cognitive theory (e.g., Bloom’s taxonomy, scaffolding theory) is cited. Strengthening this with supporting literature would make the conceptual framing more credible.

3: Dependence on extremely large teacher models. The pipeline heavily relies on Qwen3-235B-Instruct and DeepSeek-V3 (both are very big models)for data generation, verification, and judging. This raises concerns about computational cost, environmental footprint, and fairness of comparison with smaller baselines. A clear efficiency analysis is missing.

4: Weak “student” baselines. LLaMA 3.1-8B (Grattafiori et al., 2024) and small Qwen variants serve as students; these are arguably too limited for 2025 standards, reducing the significance of reported rubric-refinement gains. Stronger student baselines in the same size would provide a fairer test.

5: Marginal conceptual novelty. The methodological core largely combines known components (rejection sampling, SFT + PG training, DPO-style mapping losses). The main contribution lies in benchmark engineering and data organization rather than new learning principles.

6: Incomplete evaluation of trade-offs. The paper does not report training time, inference latency, or computational resources, leaving the cost–benefit ratio unclear. Surely it can get better results in benchmarks, but the question is will the cost worth it.

**Questions:**

What is the computational cost of training with Qwen3-235B compared to other baselines?

Could smaller “teacher” models replicate the same gains?

What theoretical or empirical justification underlies the pedagogical “hierarchy of judging skills”?

---

> ### Author Response · Authors · 2025-11-26
>
> **Q1. Domain generalization.**
>
> To evaluate performance in unseen domains, we test the models on the Science domain, including datasets such as ChemBench and ProteinLM. The results in Table D demonstrate that CompassJudger-2 maintains strong performance across objective and subjective benchmarks, as well as in unseen domains.
>
> Table D: Results on general benchmarks.
>
> | Model                            | MMLU Pro  | GPQA Diamond | AIME25    | LiveCodeBench v5 | IFEval    | ArenaHard | ChemBench | ProteinLM |
> | -------------------------------- | --------- | ------------ | --------- | ---------------- | --------- | --------- | --------- | --------- |
> | **7B Judge Models**              |           |              |           |                  |           |           |           |           |
> | Qwen2.5-7B-Instruct              | **55.43** | 34.85        | **6.67**  | 12.57            | 73.20     | 47.86     | 57.20     | 57.52     |
> | Con-J-7B-Instruct                | 44.74     | 27.27        | 3.33      | 6.59             | 54.90     | 23.49     | 45.11     | **57.63** |
> | RISE-Judge-Qwen2.5-7B            | 51.56     | 32.32        | **6.67**  | 12.57            | 44.18     | 35.99     | 57.72     | 54.24     |
> | **CompassJudger-2-7B-Instruct**  | 52.55     | **39.39**    | **6.67**  | **14.37**        | **74.49** | **53.49** | **59.37** | 54.66     |
> |                                  |           |              |           |                  |           |           |           |           |
> | **32B Judge Models**             |           |              |           |                  |           |           |           |           |
> | Qwen2.5-32B-Instruct             | 68.92     | 42.93        | 16.67     | **30.54**        | 79.85     | 70.16     | 71.75     | 61.23     |
> | RISE-Judge-Qwen2.5-32B           | 67.88     | 42.93        | 13.33     | 27.54            | 62.85     | 61.52     | **73.18** | 60.38     |
> | **CompassJudger-2-32B-Instruct** | **69.22** | **50.51**    | **23.33** | 25.15            | **79.48** | **83.31** | 72.42     | **61.65** |
>
> **Q2. & Q9. Reference**.
>
> Thank you for your advice. We incorporate Sadler’s formative assessment theory [1] and revise the corresponding part to strengthen the theoretical support.
>
>
>
> **Q3. & Q6. & Q7. & Q8. Cost analysis.**
>
> In terms of computational cost, the majority comes from training rather than inference and deployment.  We use 64 GPUs for 10 hours to train a 32B model that can be used by the community and achieves state-of-the-art performance. This level of computation aligns with what is commonly required to train baseline models.  For  larger baselines such as Skywork-Critic-Llama-3.1-70B, the training cost is even higher.
>
> Therefore, we believe the cost-benefit ratio raised by the reviewer is unnecessary. Both the baseline models and our work aim to advance the development of relevant technologies and provide the community with practical judge models.  Our provided 7B and 32B variants also ensure that users can choose a judge model that best fits their needs.
>
> As for the training data, our collected data is open-sourced for community use in training judge models, making the cost of data generation acceptable as well.  We provide more details about open-source data in Q3 in Global Response.
>
>
>
> **Q4. More student models in RubricBench.**
>
> Please refer to Q1 in Global Response for more details.
>
>
>
> **Q5. Novelty and contribution.**
>
> We believe that incorporating supervision from verified judge rewards into the SFT loss is novel. The baseline RM-R1 naively introduces RL training, which leads to instability and significant training overhead. In contrast, our approach enables more stable training while also achieving SOTA performance.
>
> Indeed, this is just an improvement we made during training. Our main contribution lies in the decomposition of judging capabilities and the development of an all-in-one model for community use. Additionally, we have provided open-source datasets for the community to train on, as mentioned in Q3 in Global Response. We hope our work could boost the development of judge models in the community.
>
> [1] Sadler, D. Royce. "Formative assessment and the design of instructional systems." *Instructional science* 18.2 (1989): 119-144.

---

### Official Review · Reviewer_Q7Ec · 2025-10-31

**Soundness:** 3
**Presentation:** 2
**Contribution:** 4
**Rating:** 8
**Confidence:** 4

**Summary:**

This paper aims to improve the general and judgement capabilities of LLM-as-a-judge evaluations systems by a specific training paradigm for judge models called CompassJudger-2. The core part of the proposed system is the training data construction for three levels of evaluations tasks: objective verification, subjective evaluation and rubric refinement. The authors carefully design curation and sampling methods to produce high-quality training data for those tasks. This paper also introduce novel benchmarks called JudgeBenchV2 and RubricBench with more scenario coverage and less GT noise. The experimental results shows that the proposed CompassJudger-2 models perform excellently on multiple judge benchmarks including the JudgeBenchV2 and RubricBench.

**Strengths:**

1 This paper proposed a comprehensive data construction pipeline for evaluation model training. The task decomposition, sampling strategy and data curation methods are logically coherent.
2 The authors makes great effort on data curation and sampling, including some dirty work such as outdated data refreshment and judgement data enhancement. I believe these data (if made public) could bring more contribution to the community than the proposed methods.
3 The introduced JudgeBenchV2 and RubricBench benchmarks bring more insight on reliable evaluation of LLM-as-a-judge systems.
4 The trained CompassJudger-2 models demonstrate excellent judgement capabilities on various scenarios.
5 The paper is generally well-written.

**Weaknesses:**

1 The training details in 3.2.1 and 4.1 are confusing. In 4.1, the paper claims the DPO and margin mapping losses are applied only on limited conditions while in 3.2.1 it seems that the mapping losses are only used for the subjective evaluation tasks. How the losses applying to different tasks and conditions are not clear.

2 Minor: typo at line 113 in Figure 2: Unconsistent-> inconsistent.

**Questions:**

The paper trains a single model for CompassJudger-2 to do a wide range of evaluation tasks. Is it possible to train separate models for specific tasks and achieve better performance? e.g. train a model with only data for objective verification tasks.

---

> ### Author Response · Authors · 2025-11-26
>
> **Q1. Explanation on Judge-PG loss.**
>
> We apply Judge-PG loss only to subjective evaluation because it is the only scenario where the reward signal aligns cleanly with a localized decision token, specifically due to the verified judge rewards presented in an A/B comparison format.
>
> In contrast, objective verification already benefits from deterministic supervision (correct or wrong), which makes policy gradient methods unnecessary. Rubric refinement, on the other hand, relies on long-horizon, sequence-level improvement that are not suitable for token-level policy gradient optimization. Therefore, subjective evaluation is the only setting in which Judge-PG loss provides stable and meaningful optimization benefits.
>
>
>
> **Q2. Typo.**
>
> Thanks for your advice. We have revised the typo in the manuscript.
>
>
>
> **Q3. Ablation of task data.**
>
> Please refer to Q2 in Global Response.

---

### Official Review · Reviewer_RzY4 · 2025-11-01

**Soundness:** 2
**Presentation:** 2
**Contribution:** 2
**Rating:** 2
**Confidence:** 3

**Summary:**

The paper introduces CompassJudger-2 (CJ-2), a generative 'all-in-one' LLM judge trained to handle three tasks: objective verification, subjective evaluation and rubric refinement. It contributes two evaluation resources, JudgerBenchV2 and RubricBench, and a training recipe that combines diverse data curation with a 'Judge-PG' mapping-loss family. In the main tables, CJ-2 reports higher scores than selected baselines across JudgeBenchV2, JudgeBench, RewardBench and VerifierBench. It also shows competitive performance on general benchmarks such as MMLU-Pro, GPQA, AIME-2025, LiveCodeBench v5, IFEval and Arena-Hard. The paper further claims that CJ-2 improves test-time 'rubric refinement' for several student models and generalises out of distribution.

**Strengths:**

* Unified evaluation system. Combining objective verification, subjective judging and rubric refinement into a single model is a clear systems contribution.

* Valuable benchamrk and dataset contributions. JudgerBenchV2 provides ranking-aware metrics and 'mix-of-judgers' references. RubricBench formalises rubric-driven test-time improvement with clear task guidelines.

* The training pipeline and ablations are well described. The paper details mapping-loss choices and includes data ablations that separate the effects of Judge-RFT and general SFT data.

**Weaknesses:**

* Limited stats. Results are reported as single numbers (without error bars, standard deviations, or significance tests). It is difficult to asses differences cannot for statistical meaningfulness. Including repeated trials and confidence intervals would strengthen claims.

* Baselines. The main tables skip several strong judge and reward-judge baselines (Skywork Critic/Reward models, GLM-judge family, InternLM reward judges, Nemotron-based judges) that rank highly on public leaderboards. While one Skywork model appears in the appendix, higher-ranked variants are still missing.

* Prompt-sensitivity analysis is limited. Only a small set of style prompts were tested, again with single number reporting. A more robust testing including paraphrases, role swaps and response-order randomisation woudl probably better assess bias and sensitivity, see [1,2].

* Sparse reproducibility details. Some important details such as the number of seeds, run-to-run variance or compute requirements are not present.

Ambiguous references. Mentions of benchmarks such as VerifyBench seem to lack attribution or explanation. Clear citations and dataset descriptions are needed to allow independent verification.

[1] Wang, P., et al. (2024). Large language models are not fair evaluators. ACL. https://aclanthology.org/2024.acl-long.511.pdf
[2] Shi, L., et al. (2024–2025). Judging the judges: A systematic study of position bias in LLM-as-a-Judge. https://arxiv.org/abs/2406.07791

**Questions:**

1. Could you report error bars, confidence intervals, and significance tests for all main results (including RewardBench and JudgeBench subsets)?

2. Several strong judge baselines (e.g. Skywork Reward/Critic models, Nemotron, InternLM reward models, GLM-judge family) are missing from the main tables. Can you include direct comparisons to these top leaderboard models to better contextualise CompassJudger-2's performance?

3. How is the “overall score” in Table 5 computed? Please provide the exact formula, normalisation, handling of ties/negative scores, and aggregation unit.

4. Current OOD claims rely on only two held-out student models. Would it be possible to expand the OOD evaluation to include more families/sizes and report variance across runs?

---

> ### Author Response · Authors · 2025-11-26
>
> **Q1. & Q4. & Q6. Error bars and more details.**
>
> Thanks for your advice. We set the temperature to 0.6 during evaluation and report the average results with error bars over three seeds.  We revise the corresponding part in the experiment section for better clarity.
>
>
>
> **Q2. & Q7. Comparison with more judge baselines.**
>
> Many of the baselines mentioned by the reviewer are scalar reward models rather than generative judge models, meaning they can only produce a single numerical score for a given question and answer, instead of a textual response. Therefore, they are unable to provide reasoning behind their decisions, and they are also limited to subjective evaluation since they cannot perform objective verification or rubric refinement tasks.
>
> Despite their shortcomings, we also include them in the Scalar Reward Models section in Table 2 in the manuscript for reference.
>
>
>
> **Q3. Prompt-sensitivity analysis.**
>
> Actually, we already randomize the response order of A/B in the prompt, which is a common practice in subjective evaluation. Furthermore, to better understand the impact of response order, we swap the initial random response order and report the performance differences under this setting, as shown in Table C.
>
> As the table shows, CompassJudger-2 is less sensitive to the prompt compared to RISE, both in terms of prompt style and response order, highlighting the robustness of our method.
>
> Table C: Prompt-sensitivity performance.
>
> | Model                           | Prompt       | Chat  | Chat Hard | Safety | Reasoning | Average |
> | ------------------------------- | ------------ | ----- | --------- | ------ | --------- | ------- |
> | RISE-Judge-Qwen2.5-Instruct-7B  | Standard     | 92.2  | 76.5      | 88     | 96.1      | 88.2    |
> |                                 | Style Prompt | +0.77 | -2.79     | -2.59  | -2.6      | -1.81   |
> |                                 | Role Swap    | +0.08 | -2.15     | -1.11  | -2.88     | -1.51   |
> | CompassJudger-2-7B-Instruct     | Standard     | 91.2  | 84.05     | 89.05  | 92.24     | 89.14   |
> |                                 | Style Prompt | +2.62 | -3.02     | -1.89  | -0.98     | -0.78   |
> |                                 | Role Swap    | -0.65 | -0.64     | -0.94  | +0.70     | -0.39   |
> | RISE-Judge-Qwen2.5-Instruct-32B | Standard     | 96.6  | 83.3      | 91.9   | 98.8      | 92.7    |
> |                                 | Style Prompt | -1.14 | -10.67    | -2.31  | -1.6      | -3.98   |
> |                                 | Role Swap    | -1.46 | -4.21     | -0.28  | +0.15     | -1.50   |
> | CompassJudger-2-32B-Instruct*   | Standard     | 91.68 | 87.5      | 90     | 96.23     | 91.35   |
> |                                 | Style Prompt | +1.16 | -0.65     | -0.27  | -0.07     | +0.05   |
> |                                 | Role Swap    | -0.56 | +1.08     | +0.41  | -0.42     | +0.13   |
>
> **Q5. References.**
>
> Thank you for your advice. We have added a clear citation and reference to VerifyBench in the revised version.
>
>
>
> **Q8. Explanation on the metric in RubricBench.**
>
> In RubricBench, we use the average relative improvement compared with the Base model as the metric to calculate the overall score of the teacher model. It measures how much, on average, a given teacher model improves over the baseline in relative percentage.
>
> Let  $s_{i,j} $ to be score of teacher model $i$ under student model $j$  and $s_{\text{base},j}$ to be score of the base student model $j$.
> Then the relative improvement for student model $j$ is defined as:
> $r_{i,j} = \frac{s_{i,j} - s_{\text{base},j}}{s_{\text{base},j}} \times 100\%.$
>
> We can then obtain the overall score for model $i$ is the  mean of these improvements across all $N$ student models:
> $Overall_i = \frac{1}{N} \sum_{j=1}^{N} r_{i,j}.$
>
> We have added the corresponding part in the Appendix for better clarity.
>
>
>
> **Q9. More student models  in RubricBench.**
>
> Please refer to Q1 in  Global Response.

---

### Author Response · Authors · 2025-11-26

# Global Response

We thank all reviewers for their valuable feedback and are encouraged by the positive comments on our contributions, including

1. Clear task decomposition.
   - A logically coherent task decomposition and clear task hierarchy. (Reviewer Q7Ec & 9wnn).

   - Provide valuable coverage of the problem space. (Reviewer 6U2P)
2. Valueable benchmark.
   - Bring more insight into reliable evaluation. (Reviewer RzY4 & Q7Ec)

   - Practical evaluation metrics and rubric refinement formalization. (Reviewer 9wnn & RzY4)
3. Solid Experiments:
   - Strong empirical results and excellent judgement capabilities. (Reviewer 9wnn & Q7Ec)

   - Well-described pipeline and ablations. (Reviewer RzY4)

In the following parts, we will first respond to the common questions raised by reviewers and then respond to the rest of the concerns of each reviewer from point to point. We believe the comments and revisions have made the paper stronger and thank all the reviewers for their help. ***Please let us know if these address your concerns and if there are any further questions that need clarification.***



**Q1. More student models in RubricBench.**

To better evaluate our model's performance on the rubric refinement task, we introduced more student models from different families into RubricBench, including the SOTA model Kimi-Linear-48B-A3B-Instruct. These student models differ in model scale (4B and 8B), architecture (dense vs. MoE), and attention mechanisms (standard vs. linear attention). The results in Table A show that CompassJudger-2 outperforms all baselines in both in-distribution and out-of-distribution scenarios, demonstrating the strong generalization capabilities of our method.

We have updated the results in the manuscript and provided a detailed explanation of the evaluation metrics we used in the appendix.



**Q2. Ablation of task data.**

To have a better understanding of how different task data components contribute to the performance, we conduct additional ablation analysis and report the results in Table B. The results show that different types of data lead to noticeable improvements on their corresponding tasks. Notably, although the all-in-one model achieves performance comparable to ablation variants on objective verification and subjective evaluation, it significantly outperforms them on RubricBench. This suggests that the rubric refinement task builds upon both objective verification and subjective evaluation, and that all three types of task data jointly contribute to the improvement.

We have revised the corresponding part in the manuscript.



**Q3. Data release.**

We open-source our benchmark and a sampled collection of the training data in the following repository to improve transparency and reproducibility. Specifically, we provide examples of training data from the three task types. In addition, we release the prompts and responses from JudgerBenchV2, as well as the prompts and student responses from RubricBench. We hope our work can help accelerate the development of judge models in the community.

Repository URL: https://anonymous.4open.science/r/CompassJudger2-Release-1D92/

---

> ### Author Response · Authors · 2025-11-26
>
> **Table A: Results on RubricBench**.
>
> | Models                            | Qwen3-8B             | Qwen3-30B-A3B        | Llama3.1-8B          | Qwen3-4B-Instruct     | Gemma3-4B-Instruct    | Kimi-Linear-48B-A3B-Instruct | Overall           |
> | --------------------------------- | -------------------- | -------------------- | -------------------- | --------------------- | --------------------- | ---------------------------- | ----------------- |
> | **Base Models**                   |                      |                      |                      |                       |                       |                              |                   |
> | Base model                        | 16.70                | 15.90                | 8.10                 | 25.90                 | 16.00                 | 30.80                        | 0.00              |
> | **7B–8B Models**                  |                      |                      |                      |                       |                       |                              |                   |
> | CompassJudger-1-7B-Instruct       | 24.20                | 26.70                | 8.15                 | 42.70                 | 20.70                 | 39.10                        | 39.11             |
> | Skywork-Critic-Llama-3.1-8B       | 23.70                | 21.50                | 7.80                 | 36.80                 | 14.40                 | 35.30                        | 20.02             |
> | RM-R1-Qwen2.5-Instruct-7B         | 26.00                | 26.60                | 6.20                 | 42.60                 | 18.60                 | 40.20                        | 35.13             |
> | RISE-Judge-Qwen2.5-Instruct-7B    | 27.20                | 24.80                | 6.60                 | 42.50                 | 20.40                 | 38.80                        | 36.32             |
> | **CompassJudger-2-7B-Instruct** * | **27.40 ± 0.50**     | **27.30 ± 0.45**     | **9.00 ± 0.30**      | 41.20 ± 0.25          | **22.20 ± 0.40**      | **40.50 ± 0.35**             | **46.03 ± 0.43**  |
> | **32B+ Models**                    |                      |                      |                      |                       |                       |                              |                   |
> | Qwen2.5-Instruct-32B              | 27.00                | 27.00                | 7.80                 | 42.00                 | 21.50                 | 39.40                        | 42.04             |
> | CompassJudger-1-32B-Instruct      | 25.40                | 26.20                | 8.00                 | 43.00                 | 20.30                 | 39.00                        | 39.19             |
> | Skywork-Critic-Llama-3.1-70B      | 23.90                | 26.20                | 8.10                 | 43.40                 | 19.50                 | 39.40                        | 37.54             |
> | RM-R1-Qwen2.5-32B                 | 27.50                | 27.30                | 9.60                 | 45.70                 | 23.00                 | 41.60                        | 51.69             |
> | RISE-Judge-Qwen2.5-32B            | 25.60                | 26.20                | 7.80                 | 42.80                 | 21.80                 | 39.60                        | 40.74             |
> | **CompassJudger-2-32B-Instruct**  | **29.70 ± 0.45**     | **28.80 ± 0.48**     | **11.50 ± 0.50**     | **47.50 ± 0.51**      | **24.80 ± 0.55**      | **42.70 ± 0.57**             | **63.00 ± 0.51**  |

---

> ### Author Response · Authors · 2025-11-26
>
> **Table B: Ablation results of task data.**
>
> | Model                       | Objective Verification | Subjective Evaluation |            |             |            | Rubric Refinement | Avg (Overall) |
> | --------------------------- | ---------------------- | --------------------- | ---------- | ----------- | ---------- | ----------------- | ------------- |
> |                             | VerifyBench            | JudgerBenchV2         | JudgeBench | RewardBench | Avg (Subj) | RubricBench       |               |
> | Objective Data Only         | 86.60                  | 57.52                 | 52.74      | 49.94       | 53.40      | 34.13             | 58.04         |
> | Subjective Data Only        | 72.50                  | 60.36                 | 62.42      | 89.11       | 70.63      | 26.97             | 56.70         |
> | Rubric Data Only            | 69.60                  | 57.75                 | 50.16      | 44.25       | 50.72      | 42.00             | 54.11         |
> | CompassJudger-2-7B-Instruct | 85.50                  | 61.25                 | 61.13      | 89.14       | 70.51      | 46.03             | 67.35         |

---

### Author Response · Authors · 2025-12-04
**Summary**

In light of recent events, we appreciate AC and PCs' significant additional effort for our community. We would like to summarize the rebuttal phase for their convenience, as we believe that all concerns have been effectively addressed.

# Strengths of our Contribution

Please refer to our Global Response. We have emphasized the logical coherency of our task decomposition, the construction of RubricBench/JudgeBench2, and the strong empirical results showing CompassJudger-2 outperforms baselines by a large margin.



# Response and Revision: `Reviewer RzY4`

`Reviewer RzY4` raised questions about error bars, baselines, and prompt sensitivity.

| Overview                                               | Response Type                                                |
| ------------------------------------------------------ | ------------------------------------------------------------ |
| **[Q1 & Q4 & Q6]** Error bars and more details.        | **Clarified & Manuscript Updated** (Set temperature to 0.6, provided 3-seed results with error bars in Table A; results updated in the manuscript) |
| **[Q2 & Q7]** Comparison with stronger judge baselines | **Addressed & Manuscript Updated** (Added Scalar Reward Models comparison in Table 2) |
| **[Q3]** Prompt sensitivity analysis                   | **Addressed** (Conducted analysis on prompts in Table C)     |
| **[Q5]** References on VerifyBench                     | **Manuscript Updated**                                       |
| **[Q8]** Explanation on the Metric in RubricBench      | **Clarified & Manuscript Updated** (Clarified the relative improvement calculation) |
| **[Q9]** More student models in RubricBench            | **Addressed & Manuscript Updated** (Added Kimi-Linear-48B-A3B-Instruct and Gemma3-4B-Instruct to benchmarks) |



# Response and Revision: `Reviewer Q7Ec`

`Reviewer Q7Ec` asked about loss function consistency and task-specific training.

| Overview                                              | Response Type                                                |
| ----------------------------------------------------- | ------------------------------------------------------------ |
| **[Q1]** Explanation on Judge-PG loss and consistency | **Clarified** (Explained the distinction between subjective evaluation task and others) |
| **[Q2]** Typo.                                        | **Manuscript Updated**                                       |
| **[Q3]** Task data ablation                           | **Addressed & Manuscript Updated** (Discussed the trade-offs and the benefits of a holistic approach for generalization) |



# Response and Revision: `Reviewer 9wnn`

`Reviewer 9wnn` focused on domain generalization, theoretical grounding, cost analysis, and novelty.

| Overview                                    | Response Type                                                |
| ------------------------------------------- | ------------------------------------------------------------ |
| **[Q1]** Domain Generalization              | **Addressed & Manuscript Updated** (Demonstrated strong performance across Science, Coding, and general domains) |
| **[Q2 & Q9]** Pedagogical theory            | **Addressed & Manuscript Updated** (Incorporated Sadler’s formative assessment theory to support the framework design) |
| **[Q3 & Q6 & Q7 & Q8]** Cost Analysis       | **Clarified** (Clarified that training cost is <10 GPU hours and the cost of data generation is acceptable) |
| **[Q4]** More student models in RubricBench | **Addressed & Manuscript Updated** (Added Kimi-Linear-48B-A3B-Instruct and Gemma3-4B-Instruct to benchmarks) |
| **[Q5]** Novelty and contributions          | **Clarified** (Highlighted the stability and benefits of the all-in-one model compared to baselines) |



# Response and Revision: `Reviewer 6U2P`

`Reviewer 6U2P` emphasized reproducibility, paper structure, and task-specific training.

| Overview                                      | Response Type                                                |
| --------------------------------------------- | ------------------------------------------------------------ |
| **[Q1]** Data release                         | **Addressed** (Provided repository link with open-sourcing data) |
| **[Q2]** Clarity and structure                | **Manuscript Updated** (Restructured Section 3.2 to clearly separate JudgeBench2 and RubricBench construction) |
| **[Q3]** Explanation on subjective evaluation | **Clarified** (Explained that ground truth label is typically obtained via human-aligned voting pipelines or via introducing subtle factual errors into the pairs) |
| **[Q4]** Task data ablation                   | **Addressed & Manuscript Updated** (Discussed the trade-offs and the benefits of a holistic approach for generalization) |
| **[Q5]** More student models in RubricBench   | **Addressed & Manuscript Updated** (Added Kimi-Linear-48B-A3B-Instruct and Gemma3-4B-Instruct to benchmarks) |

---

### Meta-Review · Area_Chair_U4b8 · 2025-12-22

**Summary:**

Reviewers question whether the work’s novelty goes beyond a well-engineered combination of existing components. The empirical evaluation omits several state-of-the-art judge models, relies on single runs without uncertainty estimates, and does not substantiate claims of generalization. Training details are sometimes unclear, particularly regarding where different mapping losses apply. The reliance on massive teacher models raises concerns about cost and the practicality of reproducing results. Key benchmarks and datasets were not available during review, limiting verification. Finally, the conceptual framing could be more rigorous, and more granular ablations, prompt-sensitivity tests, and out-of-domain evaluations are needed to support the paper’s broader claims.

**Reviewer Concerns:**

While the most concerns have been addressed, experimental results including the new ones cannot support authors' statement that their proposed judge model substantially improve the performance over baselines.

**Reviewer Scores:**

2->4,8,6,4

---

### Decision · Program_Chairs · 2026-01-26

Reject